# Vantera Mediated Quantification of Urine Citrate and Creatinine: A New Technology to Assess Risk of Nephrolithiasis

**DOI:** 10.3390/diagnostics12112606

**Published:** 2022-10-27

**Authors:** Steven P. Matyus, Justyna Wolak-Dinsmore, Erwin Garcia, Randolph M. Young, Margery A. Connelly

**Affiliations:** Labcorp, Morrisville, NC 27560, USA

**Keywords:** citrate, creatinine, nuclear magnetic resonance spectroscopy, nephrolithiasis, kidney stones, Vantera Clinical Analyzer

## Abstract

Urine citrate is often used to identify patients at risk of recurrent nephrolithiasis or kidney stones. A high-throughput assay was developed to measure urine citrate and creatinine on the Vantera^®^ Clinical Analyzer, a nuclear magnetic resonance (NMR) instrument designed for the clinical laboratory. Assay performance was evaluated and comparisons between the NMR and chemistry results were conducted. Linearity was demonstrated over a wide range of concentrations for citrate (6 and 2040 mg/L) and creatinine (2.8 and 1308 mg/dL). Intra-and inter-assay precision (%CV) ranged from 0.9 to 3.7% for citrate and 0.4 to 2.1% for creatinine. The correlation coefficients for the comparison between NMR and chemistry results were 0.98 (Y = 1.00X + 5.0) for citrate and 0.96 (Y = 0.968X + 0.97) for creatinine. The reference intervals for both analytes were confirmed. Ten endogenous and exogenous substances were tested and none were found to interfere with the assay results. In conclusion, the newly developed high-throughput NMR assay exhibited robust performance and generated results comparable to the currently utilized chemistry tests, thereby providing an alternative means to simultaneously quantify urine citrate and creatinine for clinical and research use. Furthermore, the NMR assay does not exhibit the same interference limitations as the chemistry tests and it enables multiplexing with other urine metabolite assays which saves time and costs.

## 1. Introduction

Based on a recent analysis of the National Health and Nutrition Examination Survey (NHANES) (2007–2014), the prevalence of nephrolithiasis or kidney stones in the United States was 10.1%, with the prevalence being higher in men (12.6%) than in women (7.5%) [1]. There has been an increase in the prevalence of kidney stones over the past couple of decades; an increase that coincides with an increase in the prevalence of obesity and type 2 diabetes [1,2,3,4]. Fifty percent or more of individuals who have experienced a kidney stone will likely experience another stone within 10 years of the first occurrence [4]. Therefore, identifying the causes of kidney stones as well as personalized treatment to reduce stone recurrence is of high priority.

Besides encouraging increased fluid intake, the guidelines from the American Urological Association (AUA) suggest several dietary regimens and therapeutic agents for the treatment of kidney stones [4]. However, the guidelines also state that stone composition is important for evaluating patients for proper treatment [4]. Most kidney stones are comprised of either uric acid, cystine, struvite, calcium oxalate or calcium phosphate and knowing stone composition may speak to the metabolic or genetic abnormality that led to its development as well as to the therapy that would most likely reduce stone recurrence in a particular patient [4,5]. Given that nutritional factors, such as low intake of calcium, fruits and vegetables and/or high intake of sodium, calcium and animal proteins, have been shown to be associated with stone occurrence, specific nutritional therapy, informed by both diet assessment and metabolic testing, has been shown to be more effective than generalized dietary measures in preventing stone formation [4,5,6,7]. In addition to dietary treatments, therapeutics such as potassium citrate, thiazide diuretics, allopurinol, α-mercaptopropionylglycine (tiopronin) and acetohydroxamic acid have been used to reduce the recurrence of stones of specific content [4,7,8]. Therefore, serum and urine tests for metabolic abnormalities as well as tests to determine stone composition are important for treatment decision making purposes.

AUA guidelines state that laboratory testing should include one or two 24 h urine collections obtained on a random diet and analyzed for volume, pH, calcium, oxalate, uric acid, citrate, sodium, potassium and creatinine [4]. Citrate levels are especially useful as it has been shown that low urine citrate (defined as less than the low end of the laboratory specific reference interval or <320 mg/24 h) is associated with risk of recurrent nephrolithiasis and may be used for monitoring of dietary or therapeutic treatments [4,6,9,10]. In fact, a recent publication revealed that citrate, when measured by NMR, was significantly lower in stone formers than in controls [11]. Low 24 h urine citrate is also associated with reduced renal function as measured by decreased glomerular filtration rate (GFR) and may be a biomarker of acid retention [10,12]. When a 24 h urine is not practical, however, one can obtain a random urine citrate level, which can be normalized to urine creatinine [4]. In the clinical laboratory the most common assays for quantifying urine citrate and creatinine are chemistry based, however, some of the reagents for these assays have had pandemic-related supply chain issues. For example, citrate lyase was recently difficult to obtain which led to an increase in the usual turnaround time for result reporting for the urine citrate assay. In order to avoid these supply chain issues, we developed a high-throughput, reagent-less nuclear magnetic resonance (NMR) assay that simultaneously measures citrate and creatinine in urine specimens saving both time and reagent costs. Here, we describe the development of the NMR-based assay and provide analytical validation data to show that this assay is robust and may be used for the clinical assessment of urine citrate and creatinine.

## 2. Materials and Methods

### 2.1. Quantification of Citrate and Creatinine in Urine by Peak Deconvolution

Citrate and creatinine in urine specimens were quantified using their respective NMR signals at 2.50–2.75 ppm (for citrate), and 3.07 ppm (for creatinine) as illustrated in Figure 1. Because the chemical shifts of citrate and creatinine vary with pH, TSP (3-(trimethylsilyl) propionic-2,2,3,3-d_4_ acid, sodium salt) and formate signals were used as references to identify the signals of interest. The chemical shift (distance) between TSP and formate signals is proportional to the sample pH and was used to predict the location of creatinine and citrate peaks. For creatinine, the upfield signal (3.07 ppm; labeled as “creatinine1” in Figure 1) was located based on an empirical equation relating its distance from TSP to the observed distance between TSP and formate. In addition, the expected amplitude ratio and coupling of the two creatinine signals was used to improve signal specificity ensuring that the location of the analyte signals were identified unambiguously by the assay. While both creatinine signals can be used for quantification, the upfield signal is used by the assay due to its higher intensity and by the fact that it may not be affected by differences in water signal suppression. The locations of the citrate signals were identified in a multi-step procedure. First, citrate3 (Figure 1A) was located based on an empirical equation relating its distance from TSP to the observed distance between TSP and formate as well as the observed distance between creatinine2 and TSP. This allows for a more accurate identification of the less intense citrate signals (or the signals from other compounds) especially at low analyte concentrations. Second, citrate4 was located based on its coupling with citrate3. Third, citrate2 was located based on its coupling with citrate3. Finally, citrate1 was located based on its coupling with citrate2. Expected amplitude ratios were also used to enhance citrate signal specificity.

Creatinine and citrate were then quantified using a non-negative deconvolution algorithm that decomposes the respective NMR signal regions into Lorentzian lineshapes centered at the observed analyte peak locations identified above. The deconvolution was performed iteratively to adjust the width of Lorentzian lineshapes and to incorporate Lorentzian lineshapes at the peak locations of other nearby or overlapping signals until the region was fit with high fidelity. The same width was used for all analyte and non-analyte lineshapes in a given iteration and the lineshape areas were normalized to a constant value for each iteration. While other lineshapes could be used (e.g., Gaussian), the Lorentzian lineshapes were used as these functions are related to the damped sine and cosine functions that are experimentally recorded by the spectrometer during spectral acquisition. In order to fully model the natural lineshapes of the creatinine and citrate NMR signals, five Lorentzian lineshapes were centered on each identified peak location and separated by two data points each in the deconvolution models. For citrate, each set of four Lorentzian lineshapes relating to each of the four citrate peaks were constrained in amplitude proportional to their natural relationship and normalized as a group. In order to model the baseline accurately, a linear function and a constant offset were incorporated in the deconvolution model. A Lawson-Hanson non-negative least squares fitting algorithm [13] was used for signal deconvolution. Figure 1B shows the observed creatinine signal region (red) overlaid with the mathematical model [black; a composite of analyte Lorentzians (blue), linear baseline (green), and non-analyte Lorentzians (magenta)]. Similarly, Figure 1C shows the observed citrate signal region (red) overlaid with the mathematical model (black). A non-citrate signal on the leftmost citrate region (magenta lineshapes in Figure 1C is also accounted for by the iterative deconvolution algorithm.

This deconvolution algorithm was chosen rather than simple integration of NMR peaks due to the widely varying context in which the analyte signals may occur. Figure 2 illustrates examples of the challenges created by the presence of other compounds which is particularly true for citrate. For example, the main concern for overlap or superposition with creatinine is creatine. While the pH of the buffer used ensures these signals are separated, peak overlap remains in some cases. In such situations, deconvoluting each analyte peak mitigates the significant concern for inaccuracy created by superimposed signals better than simply integrating them.

The sum of the analyte Lorentzian lineshape coefficients from the deconvolution was converted to concentration units using a factor obtained from a calibration curve (Figure 3). The calibration curves were generated by relating the coefficient sum to the known amount of creatinine or citrate standard that was spiked into a urine sample. A total of 12–13 spiked samples were tested in triplicate in order to establish standard curves for creatinine and citrate.

### 2.2. NMR Spectral Data Acquisition

Each urine sample for the citrate and creatinine assay was mixed with buffer in a 2:1 (*v*/*v*) ratio by the Vantera^®^ Clinical Analyzer (Labcorp, Morrisville, NC, USA). The aqueous buffer is composed of 1.5 M potassium phosphate dibasic (K_2_HPO_4_; Sigma-Aldrich, St. Louis, MO, USA), 38 mM sodium formate (CHNaO_2_, Sigma-Aldrich) and 2.18 mM TSP (Sigma-Aldrich). The pH of the buffer was adjusted to 6.0 ± 0.1 with 6N HCl. The prepared sample was delivered by the Vantera to the NMR flowcell for spectral acquisition and processing. A pulse-acquire experiment using a 90° flip angle was implemented by the Vantera which was equipped with a 400 MHz (9.4 T) Agilent^®^ spectrometer (Agilent Technologies, Santa Clara, CA, USA). A WET [14] sequence was applied for 68 ms to attenuate the water signal. The 1D ^1^H NMR spectra were acquired at 47 °C for the purpose of compatibility with other assays on Vantera. Additional acquisition parameters include: spectral width = 4496.4 Hz, steady state scans = 4, direct detection time = 2.0 s, relaxation between scans = 1.95 s, number of scans = 12. The total time to collect each NMR spectrum is 64 s. The free-induction decay signal was zero-filled to 16,384 pairs of real and imaginary data points and was multiplied by an exponential window function corresponding to a line-broadening of 0.5 Hz prior to Fourier transformation (FT). After FT, the spectrum was corrected for phase and baseline errors. A processed spectrum with relevant peak assignments is shown in Figure 1. Simultaneous results generation for both citrate and creatinine follows the NMR spectrum collection while the sample was being replaced in the flow cell by the next sample, resulting in a sample to results throughput of less than 2 min.

### 2.3. Analytical Validation of the Urine Citrate and Creatinine Assay

Imprecision was determined according to Clinical and Laboratory Standards Institute (CLSI) [15]. Several urine pools were prepared by combining selected samples from specimens collected in urine collection containers. Three pools with low, intermediate and high citrate and creatinine concentrations were chosen to evaluate assay imprecision (mean, standard deviation, and %CV). Intra-assay (within-run) precision for each pool was determined from 20 observations (1 pool × 1 run × 20 replicates = 20). Inter-assay (within-laboratory or between runs) precision was determined from 74–80 observations of each pool (20 days × 2 runs × 2 replicates = 80). Mean, standard deviation (SD) and % coefficients of variation (%CV) were calculated using EP Evaluator^®^ (software, tool; v11.3.0.23, Data Innovations LLC, Colchester, VT, USA). The acceptance criteria for citrate and creatinine assay imprecision were predetermined to be 10% and 12%, respectively, based on published biological variation databases [16,17].

Linearity was evaluated according to CLSI guidelines [18] using serially mixed source pools with intermediate (155 mg/L citrate, 60 mg/dL creatinine), and high analyte concentrations (2235 mg/L citrate, 1435 mg/dL creatinine) in order to cover a wide concentration range. Deionized water was used to dilute the intermediate pool in order to generate the low concentrations. The pools were tested in quadruplicate on a single instrument in one day. Assay linearity was assessed by linear and higher order polynomial regression of the test results of the prepared mixtures compared to the expected concentrations using EP Evaluator^®^ (software, tool; v11.3.0.23, Data Innovations LLC, Colchester, VT, USA). Acceptance criteria for the linearity data was defined as allowable nonlinearity (or %bias) of 3.5% for citrate and 8.6% for creatinine, and a slope between 0.90–1.10. To calculate the limit of blank (LOB) and limit of detection (LOD), five deionized water samples and five low concentration samples were tested, respectively. For the lower limit of quantification (LLOQ), eight urine samples were used to determine the LLOQ. Four (4) replicates per pool per day were tested for 3 days according to CLSI guidelines [19] as previously described [20,21]. To determine the LLOQ, the experimental points (%CV versus mean concentration) were fitted to a power function. The concentration of the analyte corresponding to the predetermined imprecision expressed as %CV was interpolated as the LOQ. The imprecision limits for citrate and creatinine were predetermined to be 15% and 18%, respectively [16,17].

Fresh urine specimens from donors were used to assess the stability of citrate and creatinine at the following temperatures: room temperature (20–25 °C), refrigerated at 2–8 °C and frozen at −20 °C and <−70 °C. Specimens were also tested for up to five freeze–thaw cycles. Citrate or creatinine, in urine specimens, was deemed stable if the difference between the result at time 0 and the following time point was less than 10%.

### 2.4. Comparison of Results from the NMR and Chemistry Tests

To enable the method comparison study, de-identified residual urine specimens were collected from the clinical laboratory (Labcorp, Burlington, NC, USA). Specimens were collected in urine specimen containers (e.g., random urine transport tube or urine Monovette^®^ (tube, system) with hydrochloric acid as a pH stabilizer for a final pH between 1 and 3) as per manufacturer’s instructions. In lieu of added hydrochloric acid, specimens could be frozen to preserve the stability of the analytes. All procedures were cleared by a local Institutional Review Board (IRB).

Method comparison studies were performed comparing results from the NMR-based assay with citrate and creatinine concentrations determined using the citrate lyase and creatinase-based enzymatic assays briefly described here, as per CLSI guidelines [22]. For the citrate assay, citrate is converted to oxaloacetate in a reaction catalyzed by citrate lyase. In the presence of malate dehydrogenase and lactate dehydrogenase, oxaloacetate and its decarboxylation product pyruvate are reduced by NADH to L-malate and L-lactate, respectively. Citrate is quantified by measuring the change in absorbance of NADH at a wavelength of 340 nm, as per manufacturer’s instructions. For the creatinine assay, creatinine reacts with alkaline picrate to form a color that can be measured at 500 nm, as per manufacturer’s instructions. The rate of color development between 0.5 and 2.0 min is proportional to the amount of creatinine in the sample. Non-creatinine chromogens do not react appreciably during this interval thus sample dialysates or protein-free filtrates are not necessary. Samples were run on Beckman Coulter^®^ AU680 instruments. Results were reported in mg/L for citrate and mg/dL for creatinine. De-identified residual urine specimens were collected after the chemistry assays were performed and taken to the NMR lab for subsequent testing. Data analysis was performed and outliers were detected and removed using EP Evaluator^®^ (software, tool; v11.3.0.23, Data Innovations LLC, Colchester, VT, USA).

### 2.5. Reference Intervals for Citrate and Creatinine in Urine

The reference intervals for citrate and creatinine were confirmed using adult urine specimens (*n* = 310) randomly collected from specimens that were sent to the clinical laboratory for analysis. Citrate and creatinine were quantified by employing the newly developed NMR assay as described above. The reference intervals were determined from the 2.5th and 97.5th percentiles using EP Evaluator^®^.

### 2.6. Evaluation for Interfering Substances

Substances (*n* = 10) were tested in vitro for potential interference with results produced by the urine citrate and creatinine assay (3 endogenous and 7 exogenous substances). Pooled urine with citrate concentrations between 228.6–885.7 mg/L and creatinine concentrations between 57.1–128.6 mg/dL were used to generate the substance interference data during the initial screening. Substances that showed interference during the initial screening were tested in a dose response fashion according to CLSI guidelines [23]. For acetic acid and boric acid, which can be used as preservatives in urine, the recommendation is to test 5 times the suggested concentration. For acetic acid, the recommended concentration of 0.5% to 2.5% was tested and for boric acid the recommended concentration of 1% to 5% was tested. The highest concentration tested where there was no interference with citrate and creatinine results was defined as <10% bias for citrate and <12.9% bias for creatinine [16,17].

## 3. Results

### 3.1. Quantification of Citrate and Creatinine Using an NMR-Based Algorithm

The concentration of citrate and creatinine were determined by modeling their respective signal peaks elicited by the methylene and methyl group protons captured within the spectrum of urine (Figure 1). The peaks were modeled to their component lineshapes and the citrate and creatinine concentrations were determined by converting the modeled deconvolution coefficients of their distinctive signal peaks into concentration units (mg/L for citrate; mg/dL for creatinine). With standardized NMR spectra, such as those produced by the Vantera^®^ Clinical Analyzer (Labcorp, Morrisville, NC, USA)., the amplitudes of the signal peaks are linearly related to their respective analyte concentrations [24]. The signal peaks for citrate and creatinine were transformed into concentrations using conversion factors determined from standard curves with known citrate and creatinine concentrations (Figure 3).

### 3.2. Linearity and Sensitivity of Citrate and Creatinine Measurements

Assay results were assessed for linearity using regression analyses of the measured versus assigned urine citrate and creatinine concentrations (Figure 4). The citrate results were linear over a range of 6 to 2040 mg/L. The equation for the best line for citrate was determined to be Y = 1.01X − 0.18. None of the polynomial fits were statistically better than the linear fit at the 5% significance level. For citrate, the limit of blank (LOB), analytical sensitivity or limit of detection (LOD), and functional sensitivity or lower limit of quantitation (LLOQ) were determined to be: 6, 11 and 18 mg/L, respectively. The creatinine results were linear over a range of 2.8 to 1308 mg/dL. The equation for the best line for creatinine was determined to be Y = 1.00X − 0.24. The 3rd order polynomial fit was statistically better than the linear fit at the 5% significance level. For creatinine, the LOB, LOD, and LLOQ were determined to be: 0.4, 1.3 and 2.8 mg/dL, respectively.

### 3.3. Assay Performance and Stability of Citrate and Creatinine in Urine

Intra-assay (within-run) and inter-assay (within-lab) precision were evaluated for the NMR assay using urine pools with low, intermediate, and high concentrations of citrate and creatinine. The results are summarized in Table 1. The %CV ranged from 0.9 to 2.9% for intra-assay, and 2.9 to 3.7% for inter-assay precision for the measurement of citrate. The %CV ranged from 0.4 to 0.7% for intra-assay, and 1.5 to 2.1% for inter-assay precision for the measurement of creatinine.

The stability of citrate and creatinine was evaluated after urine samples were stored at different temperatures and after multiple freeze–thaw cycles (Table 2). Both citrate and creatinine were stable for up to 29 days when stored at ambient room temperature and when refrigerated. Both were stable for up to 49 days when frozen at <−70 °C (Table 2). In addition, no significant change in either citrate or creatinine results were observed after five freeze–thaw cycles. Stability may extend beyond the time points tested in the current study especially for frozen samples.

### 3.4. Comparison of Citrate and Creatinine Results from the NMR-and Chemistry-Based Assays

A method comparison study was performed to compare NMR-based citrate and creatinine test results with results generated using the chemistry-based assays. Deming regression analysis of the citrate results (*n* = 297) from both assays produced a correlation coefficient of 0.977, and a slope and intercept of 0.971 and 4.7, respectively (Figure 5A). The Bias plot for citrate revealed no systematic bias between the results of the two assays (mean bias= −3.0%) (Figure 5B). For creatinine, the Deming regression analysis (*n* = 306) from both assays produced a correlation coefficient of 0.960, and a slope and intercept of 0.968 and 0.97, respectively (Figure 5C). The Bias plot for creatinine revealed no systematic bias between the results of the two assays (mean bias= −1.4%) (Figure 5D).

### 3.5. Reference Intervals in Apparently Healthy Subjects

The reference intervals for the citrate and creatinine assay were determined in urine specimens obtained from the clinical laboratory (*n* = 310). The mean citrate concentration in this population was 245 ± 203 mg/L, the reference interval was 15–854 mg/L and the range of citrate values was 2–1458 mg/L. The mean creatinine concentration in this population was 74.5 ± 41.8 mg/dL, the reference interval was 18.7–174.5 mg/dL and the range of creatinine values was 8.9–223.4 mg/dL. These results confirm reference intervals that were previously determined using the chemistry assays (available on Labcorp test menu).

### 3.6. In Vitro Testing of Substances for Potential Interference with Test Results

A total of 3 endogenous (e.g., urea, uric acid and albumin) and 7 exogenous (acetaminophen, acetic acid, acetylsalicylic acid, ascorbic acid or vitamin C, boric acid, ibuprofen and naproxen sodium) substances were tested for potential interference with accurate citrate and creatinine assay results. Table 3 shows the highest substance concentrations tested that did not elicit interference. These results revealed that none of these substances interfered with the generation of citrate and creatinine test results within the concentrations at which they naturally occur (endogenous) or at their therapeutic concentrations (exogenous). This includes acetic and boric acid which can be used in lieu of hydrochloric acid as a preservative in urine specimens. While these acids interfere with the chemistry-based assays, concentrations higher than the concentration usually used for preserving urine analytes at the highest concentrations suggested for use (1000 mg/dL boric acid and 1515 mg/dL acetic acid) did not interfere with the NMR-measured citrate and creatinine results.

## 4. Discussion

In the clinical laboratory the most common tests for quantifying urine citrate and creatinine are chemistry assays, however, some of the reagents for these tests have had pandemic-related supply chain issues. For example, citrate lyase was recently difficult to obtain which affected the usual turnaround time for result reporting for the urine citrate assay. In order to avoid these supply chain issues, we developed a high-throughput, reagent-less NMR assay to simultaneously measure both citrate and creatinine concentrations in urine specimens. The comparison of the results from the NMR-based assays for urine citrate and creatinine revealed high correlation coefficients (0.98 and 0.96, respectively), small intercepts (4.7 and 0.97, respectively) and slopes of 0.971 and 0.968, respectively, suggesting that the NMR-based results can substitute for the chemistry-based results. Precision studies showed that the NMR-based assays had good precision (%CV for both assays ≤3.7% as shown in Table 1) and measured citrate and creatinine accurately (as shown in Figure 5). Finally, while urine preservatives such as acetic acid and boric acid are listed as limitations for the chemistry based assays, these interferences were not found to be a limitation for the NMR-based assay. Therefore, the NMR-based assay has performance characteristics that would allow it to be used for clinical decision-making purposes.

Besides having good performance characteristics for quantifying citrate and creatinine, the NMR-based assay has several benefits over the chemistry-based assays. Some of these benefits are not having the same limitations as the chemistry assays, e.g., interference by use of acetic acid or boric acid as preservatives. The NMR assay is reagent-less and therefore not dependent on reagents such as citrate lyase that have occasionally had supply chain issues. Moreover, NMR is high-throughput and there is no manipulation of the sample before testing (e.g., dilution of the sample with diluent buffer occurs on board the instrument), therefore the turn-around-time for testing and reporting of results is <2 min. Furthermore, the NMR-based assay provides results for both citrate and creatinine simultaneously from the same spectral acquisition for the same specimen. This highlights one of the benefits of NMR, that several analytes can be quantified simultaneously thereby significantly reducing time, resources, and the costs for testing. Another example of this is the current serum/plasma assay for lipoprotein analysis which enables the simultaneous quantification of lipoprotein particles, small molecule metabolites, and GlycA, an NMR-specific marker of systemic inflammation [20,25,26,27,28,29,30,31,32]. The fact that NMR assays are high-throughput and easy to use makes them amenable for use in testing samples from large observational and interventional clinical studies as well. While the current NMR assay is able to quantify citrate and creatinine in urine specimens, future application of the technology may also include quantification of cystine and uric acid, which would allow for a more extensive analysis of risk of kidney stone formation. Taken together, the new NMR-based assay offers a practical alternative method to quantify metabolites in urine.

## 5. Conclusions

The newly developed high-throughput NMR assay exhibited good performance and generates results comparable to the currently utilized chemistry tests and provides an alternative means to simultaneously quantify urine citrate and creatinine for clinical and research use.

## 6. Patents

SPM, JWD and EG are listed as inventors on a patent for this assay.

## Figures and Tables

**Figure 1 diagnostics-12-02606-f001:**
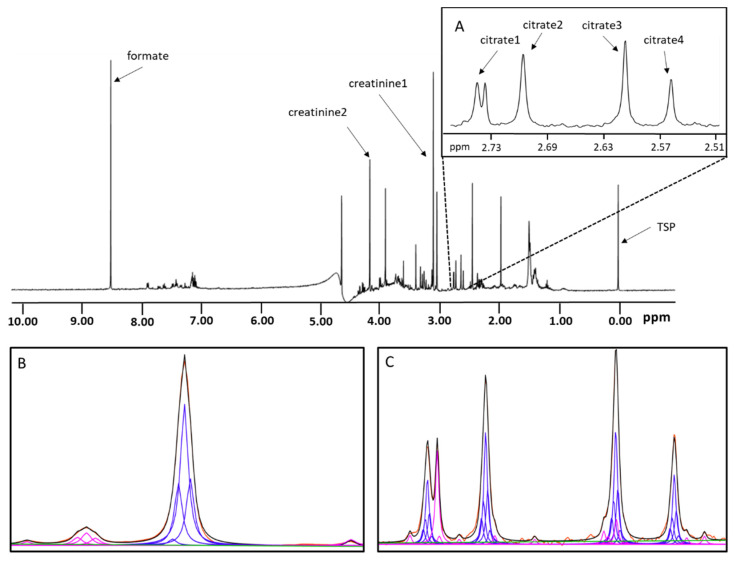
Representative 1D ^1^H NMR spectrum for urine using a pulse-acquire experiment with WET solvent suppression. Resonance assignment for relevant signals are shown. (**A**): spectral expansion showing the four citrate signals. (**B**): Overlay of creatinine observed signal (red) and the mathematical model (black), along with analyte Lorentzian (blue), non-analyte Lorentzian (magenta) and linear baseline (green) components. (**C**): The citrate observed signals (red) are overlaid with the mathematical fit (black). The analyte Lorentzian lineshapes (blue), non-analyte Lorentzian lineshapes (magenta) and linear baseline (green) are also shown.

**Figure 2 diagnostics-12-02606-f002:**
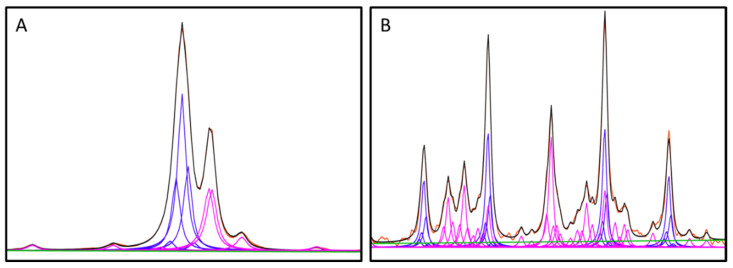
Overlays of observed NMR signal (red) and the mathematical model (black), along with analyte Lorentzian (blue), non-analyte Lorentzian (magenta) and linear baseline (green) components resulting from deconvolution of creatinine and citrate regions. (**A**): Creatinine region with nearby or overlapping signals. (**B**): Citrate region with nearby or overlapping signals.

**Figure 3 diagnostics-12-02606-f003:**
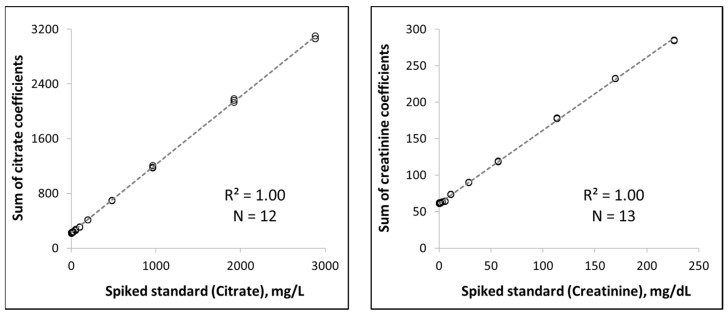
Standard curves for the determination of citrate and creatinine concentrations.

**Figure 4 diagnostics-12-02606-f004:**
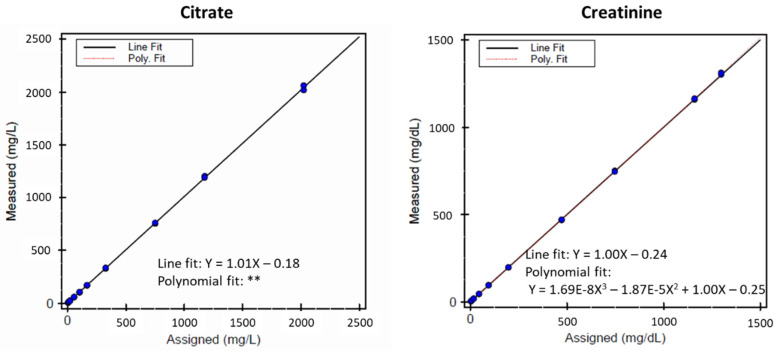
Linearity of NMR-measured versus assigned values of citrate (**left**) and creatinine (**right**) in the urine citrate and creatinine assay. ** After regression, no 2nd/3rd order polynomial fit was statistically better than the linear fit at the 5% significance level. Thus, only the linear fit is shown.

**Figure 5 diagnostics-12-02606-f005:**
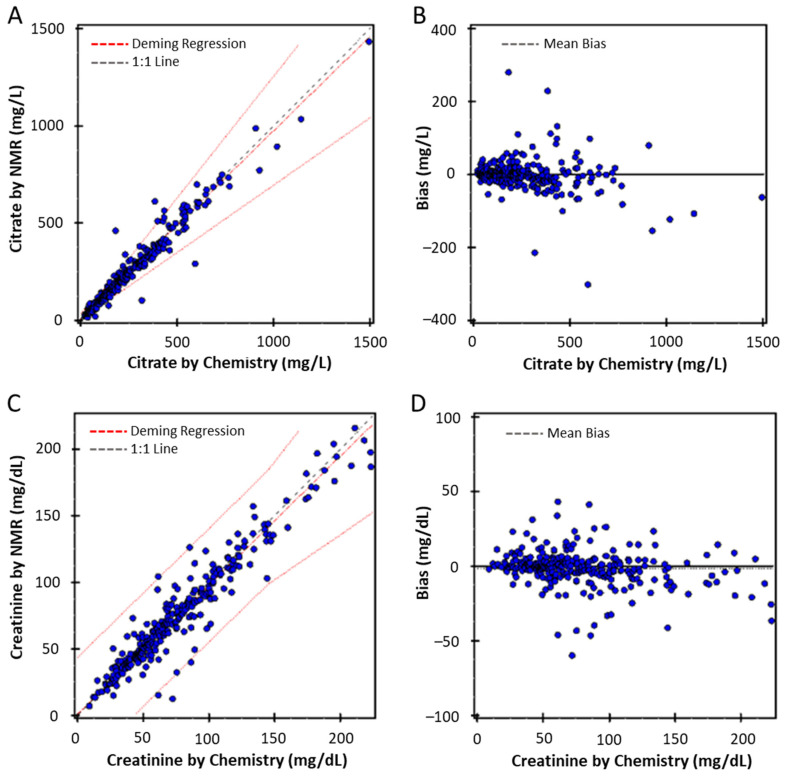
Deming regression comparison between chemistry and NMR measured citrate (**A**) and creatinine (**C**). Bland–Altman plots of the residuals for citrate (**B**) and creatinine (**D**). The limits of agreement (LOAs) are depicted as dotted blue lines and 0% bias is a solid grey line.

**Table 1 diagnostics-12-02606-t001:** Intra- and Inter-assay imprecision for citrate and creatinine measured in urine.

	Citrate (mg/L)			Creatinine (mg/dL)		
	Low	Intermediate	High	Low	Intermediate	High
	**Intra-assay (within-run) ^a^**					
Mean	94.4	248.8	482.1	60.3	114.3	173.0
SD	2.8	2.3	3.5	0.6	0.7	1.2
%CV	3.0	0.9	0.7	0.9	0.6	0.7
	**Inter-assay (within-lab) ^b^**					
Mean	91.9	246.6	480.1	59.2	113.5	173.6
SD	3.4	7.8	12.9	1.4	2.8	4.3
%CV	3.7	3.2	2.7	2.3	2.4	2.5

^a^ Based on 1 run of 20 tests (*n* = 20), ^b^ Based on CLSI EP5-A2 tested using 2 runs per day in duplicate for 20 days (*n* = 80).

**Table 2 diagnostics-12-02606-t002:** Summary of citrate and creatinine stability in urine specimens.

Storage Condition	Citrate	Creatinine
Controlled Room Temperature, 20 to 25 °C	Up to 29 days	Up to 29 days
Refrigerated, 2 to 8 °C	Up to 29 days	Up to 29 days
Frozen, <−70 °C	Up to 49 days	Up to 49 days
Freeze–thaw, −70 °C	5×	5×

**Table 3 diagnostics-12-02606-t003:** Results of interference testing showing highest concentration of substance tested that did not interfere with citrate or creatinine assay results.

Substance	Drug Name	Concentration (mg/dL)
Urea	─	263.9
Uric acid	─	23.5
Protein (albumin)	─	175.0
Acetaminophen	Tylenol	21.8
Acetic acid	Preservative	2500
Acetylsalicylic acid	Aspirin	66.4
Ascorbic acid	Vitamin C	6.0
Boric acid	Preservative	1250 *
Ibuprofen sodium salt	Advil	59.0
Naproxen sodium	Aleve	56.1

* Interferes with assay results at this concentration, which is higher than the highest concentration of boric acid when used as a preservative.

## Data Availability

Data are available upon request.

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
