# Peer review of "Vantera Mediated Quantification of Urine Citrate and Creatinine: A New Technology to Assess Risk of Nephrolithiasis"

_diagnostics, 2022, doi:10.3390/diagnostics12112606_

Round 1

Reviewer 1 Report

The authors present an interesting development for measuring urine citrate and creatinine with on the Vantera® Clinical Analyzer (a nuclear magnetic resonance (NMR) instrument designed for the clinical laboratory). This work provides measurements for certain concentration range of urine citrate and urine creatinine. They claim that, the newly developed high-throughput NMR assay providing an alternative way to simultaneously quantify urine citrate and creatinine for clinical and research use. While I find their experimental approach is interesting, the motivation of this work is weak. They said to avoid chemistry supply chain issues, they developed a high-throughput, reagentless nuclear magnetic resonance (NMR) assay to simultaneously measure both citrate and creatinine in urine specimens. Is the reason of this work, what are the other advantages; fast, cheap, practical etc.?

The methods and results have lots of information which have lacks of clarity. So, I don't feel that they have translated their observations well in the result and discussion part. And I have doubt about samples and storage of these samples. Finally, most figures should be plotted again.

Specific comments:

1-Lines between 39-41, it says Most kidney stones are comprised of either uric acid, cystine or struvite and knowing stone composition may speak to the metabolic or genetic abnormality that… It should be mentioned the other types of stones as well, such as calcium oxalate, etc.

2.Line 43, such as low intake of calcium causes stone formation. But high intake of calcium also associated with stone occurrence.

3. Line 54-56, It says Citrate levels are especially useful as it has been shown that low urine citrate is associated with risk of recurrent nephrolithiasis and may be used for monitoring of dietary or therapeutic treatments. To make your motivation strong, could you please explain which concentrations range of creatinine and citrate sign of the nephrolithiasis?

4. Line 83, …in water signal suppression. The locations of the citrate signals were… What is ph value of water?

5. Line 92-93, Did you try other models to compare with accuracy of Lorentzian line shapes. Could you please mention of other possible models and advantage of this model?

6. Line 62-66, it is written that For example, citrate lyase was recently difficult to obtain which led to an increase in the usual turn around time for result reporting for the urine citrate assay. In order to avoid these supply chain issues, we developed a high-throughput, reagent-less nuclear magnetic resonance (NMR) assay to simultaneously measure both citrate and creatinine in urine specimens. Is this only reason?

7. Line 77, Figure 1), should be Figure 1

8. Figure 1. B & C, the peaks overlap and difficult to distinguish clack red and green. Please change width or shape to make them visible.

9. Line 166, Could you please specify value of high concentrations of citrate or creatinine?

10. Figure 3-Citrate, it is written that Poly fit, **No 2nd/3rd order polynomial fit was statistically better than the linear fit at the 5% significance level.

It is not significant way to mention poly fit in graph without plotting and then saying that it is statistically better.

11. I recommend to replot Figure 1.B&C, Figure 3 and Figure 4 with higher quality and making color, width, symbols better.

12. Line 279-281, it is written that In addition, no significant change in either citrate or creatinine results were observed after five freeze-thaw cycles. Stability may extend beyond the time points tested in the current study. I do not understand which results prove that there is no significant change?

13. Line 347-348, Precision studies showed that the NMR-based assays had good precision (%CV for 347 both assays ≤3.7%) and measured citrate and creatinine accurately (data not shown). Why data not shown?

Reviewer 2 Report

In the manuscript entitled “Vantera Mediated Quantification of Urine Citrate and Creatinine: A New Technology to Assess Risk of Nephrolithiasis” the authors evaluated whether the Vantera® Clinical Analyzer, a nuclear magnetic resonance (NMR) instrument could be used to measure citrate when compared to a well-known chemical assay in ~ 300 healthy individuals. Creatinine measurement was also performed as a control. Their main findings included an important linearity between assays with excellent intra- and inter-assay precision. The study is clinically relevant and contributes to knowledge in the field.      

Minor comment

1.    Could you provide the results of the interference of other substances, such as calcium, oxalate, and sodium, in citrate and creatinine measurements, as these parameters are frequently altered in individuals with lithiasis?  

Reviewer 3 Report

In this paper authors developed a high throughput, “reagent-less” NMR assay to simultaneously measure citrate and creatinine concentrations in urine specimens. I think that introducing “reagent-less” methods into clinical labs is very important improvement in clinical analysis and I am truly supportive of this study.

Overall, the paper is well written, and methods used were well documented, the introduction may need some more references (see below), and a rationale why were these 2 metabolites selected.

The only exception is the analysis of NMR spectra where proprietary method has been used. This is a major comment/issue I have. I find using proprietary methods in scientific papers unacceptable, and unless authors fully describe their method, despite overall positive impression I cannot recommend paper for publication. First, in a process of peer review we have to know the specific methodology. Second, how could someone from scientific community, other researchers interested in proposed method reproduce results at a different institution, or further apply the method? If the authors still insist on proprietary method, then the scientific literature is not the place for this study.

Below are related questions which if answered could shed more light on the analysis method used:

1.     In Fig. 1 authors show that experimental NMR spectral lines could be fit by superposition of Lorentzian lines, 3 for creatinine and 5 for citrate:

a.       How was the number of Lorentzian lines determined (3 vs. 5)?

b.       How were the positions of lines determined: from known long range J coupling, or from the best fit of the pattern, or some other way?

c.       How were intensities of the lines determined: J-coupling or best fit of the pattern?

d.       How were widths of the lines determined: from line broadening or best fit?

2.       How were the concentrations determined from simulated/fitted patterns?

3.       Normally, in the absence of serious overlap (like here) one just integrates phase- and baseline-corrected lines. Why was it not done here?

4.       What is “the amplitude” of the signal? Presumably, it is a peak intensity (area); if it is relative, what is the reference, and if it is absolute, what is the unit?

5.       The authors say they use the expected amplitude ratio of the two creatinine signals for unambiguous identification, and only the creatinine1 signal is used for quantification. What if creatinine1 peak overlaps with an exogenous substance, and the ratio of the two signals is not as expected? Were any corrections applied?

6.       Why were the spectra acquired at 47°C?

Comments about figures:

The signals in Figure 1, panel C should be increased, it is difficult to see the all the lines and colors.

The resolution of the spectrum in the Panel A could be improved.

Other minor remarks:

Authors are encouraged to review and cite in their introduction recent relevant work not currently cited. These 2 papers for sure: PMIDs 35933132 and 35853716 (from this paper review and potentially cite references 39 and 42 about citrate).

EP Evaluator: authors mentioned detail information about version number used, and the company that made the software at the bottom of the page 5, line 216. This should have been done earlier, at the first mention of this software.

Line 75: In TSP (3-(trimethylsilyl) propionic-2,2,3,3-d4 acid, d4 should be d4

Line 135: K2HPO4 should be K2HPO4

Line 136: CHNaO2 should be CHNaO2

Line 140: msec should be ms

Line 143 : sec should be s

Line 141 (and 109): 1D 1H NMR should be 1D 1H NMR

Line 348: data not shown should never be put in the discussion, potentially in results if journal style allows.

Round 2

Reviewer 1 Report

Dear Editor and authors,

I believe the manuscript has been improved well.

Kind regards,

Author Response

Reviewer #1 comment: 

I believe the manuscript has been improved well.

Reviewer #1 Response:

Thank you for the kind comment and for taking the time to review our manuscript.

Reviewer 3 Report

This revision significantly improved and I would just add one final minimal comment. In the introduction, the authors added one reference I suggested (35933132), but did not the second one (35853716). I would suggest that adding the second reference would be beneficial as rationale why they studied citrate, at least with one sentence. As authors can find, citrate assessed by urine chemistries did not differ between controls and stone formers, but it did differ when NMR was done.

Author Response

Reviewer #3 comment: 

This revision significantly improved and I would just add one final minimal comment. In the introduction, the authors added one reference I suggested (35933132), but did not the second one (35853716). I would suggest that adding the second reference would be beneficial as rationale why they studied citrate, at least with one sentence. As authors can find, citrate assessed by urine chemistries did not differ between controls and stone formers, but it did differ when NMR was done.

Reviewer #3 Response:

Thank you for catching our oversight.  We have now added the Thongprayoon et al publication to the introduction as well as references 39 and 42 from this paper.  We have also added the following statement to the introduction section: “In fact, a recent publication revealed that citrate, when measured by NMR, was significantly lower in stone formers than in controls. ” (see lines 58-60)